# Intrinsic Behavioral Variability Facilitates Flexible Representations: A Neuromotor Developmental Perspective

## Abstract

Dynamic human movement necessitates a dynamic representation of the body. The mechanisms underlying the initiation, development, and maintenance of such representations can provide a biological perspective to developing more flexible representations within computational agents. Taking inspiration from the prenatal twitches shown to initiate the human neuromotor representation, we question how these same twitches, present throughout development, may also facilitate subsequent motor adaptation. Across three experiments, we examine the influence twitches, as a form of intrinsic behavioral variability, may have in facilitating motor adaptation to novel situations. In a series of simulated reaching tasks, we trained agents to reach targets while overcoming behavioral, physiological, and neurological changes. Overall, we found evidence that agents exposed to intermittent behavioral variability outperformed their counterparts, showing greater neural weight variability, indicative of greater exploration. Taken together, this work provides a biologically plausible computational framework for flexible representation development.

## 1 Introduction

Dynamic representations of the human body are essential for adapting to the dynamic internal changes and external stimuli we experience every day. Our bodies change dramatically as we develop, growing and lengthening through adolescence, where at the same time, we are tasked with learning a multitude of novel skills (i.e., walking, holding, driving). Similarly, as we progress through adulthood, we again may face changes and challenges, albeit typical bodily shrinkage or, for some, the dramatic loss of a limb or degradation of the mind. Despite this, humans show an affinity for adapting to these changes. Much of this may be due, in part, to the dynamic representations we build about our bodies throughout development (Wang et al., 2022). How these representations form and change in response to novel stimuli, however, is still unclear.

Traditionally, the human motor mapping was thought to be a static two-dimensional somatotopic representation of the body, mapping specific regions of the body (e.g., body, leg, face, hips) to specific parts of the cortical folds of the motor cortex (Penfield & Boldrey, 1937). An abundance of developmental biology evidence has supported this dogma, pointing to myoclonic twitches as the catalyst for this representation (Blumberg et al., 2013b; Chehade & Gharbawie, 2023; Gentilucci et al., 1989; Grodd et al., 2001; Inácio et al., 2016; Petersson et al., 2003; Sokoloff et al., 2020). During human gestation, as motorneurons differentiate, innervating the skeletal muscles, these neurons begin to spontaneously fire, cascading rhythmic action potentials along nearby muscle cells (Blumberg et al., 2013a; Thomason et al., 2018). Through mechanisms unknown, these myoclonic twitches, a form of spontaneous muscle activation (SMA), develop a spatiotemporal organization that can be seen as clustered twitches within body areas of ethological significance (Blumberg et al., 2013a). These same twitches, though lessened in frequency and intensity postnatally, persist throughout the lifespan, apparent during periods of rest. How these twitches play a subsequent role in our motor representation postnatally remains a topic of debate (Sokoloff et al., 2020).

Despite this evidence for a static, spatially-dependent, somatotopic motor mapping, research has also begun to show evidence that the mapping is more dynamic than previously assumed. In a follow-

up to the foundational study supporting a somatotopic representation of the body, evidence showed that the same regions of the motor cortex produced multi-jointed and stereotype actions when stimulated for ecologically valid periods, contrasting the single-jointed actions seen previously (Graziano, 2016). Similar multi-jointed actions were found longitudinally as well. In a series of studies observing infant-rat behaviors and their neural correlates, evidence showed developmental changes in the mapping between neurons and actions. Specific neurons in the rats that elicited twitch-like movements at eight to ten weeks (8-10) old, mirroring a somatotopic representation, now, by twelve weeks (12), elicited action-oriented movements (Dooley & Blumberg, 2018). Similar sensory representational changes were noted as well in human infants, several months postnatal (Dall'Orso et al., 2021). This culminating evidence has led researchers to view our motor representation as more dynamic than originally presumed, developing initially as somatotopic mappings of the body followed by a more ethological mapping of the body shortly after birth. The mechanistic underpinnings that could initialize, update, and maintain such a dynamic representation, however, remain a topic of contention.

While it has become more widely agreed upon that the human motor mapping represents both a somatotopic and ethological representation of the body, there remains a lack of understanding as to how the representational developmental trajectory influences subsequent behavioral adaptations to novel stimuli. Emerging evidence has led us to investigate the benefit of intrinsic behavioral variability (IBV), such as SMAs, in representational maintenance and flexibility for skill learning (Kuebrich & Sober, 2015; Leitão & Gahr, 2024; Sokoloff et al., 2020). Just as SMAs have been argued to initiate the somatotopic motor representation prenatally, we hope to explore how injecting variable behavior into a system throughout training may also serve to improve representational organization and subsequent performance. As such, we hypothesize that IBV facilitates a flexible representation of an agent's behavioral repertoire and self, encouraging adaptation throughout training. To address this hypothesis, we explored training agents in various adaptation tasks, injecting behavioral variability in three (3) different intervals throughout training:

## 1.1 TRAINING HYPOTHESIS [H0]

This agent will be given no IBV, building on neurological and computational evidence by Graziano and colleagues showing that what initially appeared globally as a somatotopic representation of the body, may simply be an amalgam of ethological actions (Aflalo & Graziano, 2006; Graziano, 2016; Meier et al., 2008). As such, this hypothesis argues that learning across a diverse array of environments will provide sufficient training for later adaptation.

## 1.2 PRE-TRAINING IBV HYPOTHESIS [H1]

This agent will simulate IBV before training, building on neurological theories that the human motor representation is initialized somatotopically prenatally via SMAs but is subsequently overridden by ethological actions. In contrast to the H0 Theory, this theory argues that flexible representations must be initialized by variable behaviors, as evidenced by Blumberg and colleagues (Thomason et al., 2018).

## 1.3 INTERMITTENT IBV HYPOTHESIS [H2]

This agent will simulate variable behaviors before and throughout training, building on arguments that post-natal SMAs may play a role similar to that of their prenatal counterparts (Blumberg et al., 2013b; Sokoloff et al., 2020). As such, this hypothesis argues that in addition to initializing the representation, the interspersed variable behaviors improve a representation's flexibility by preventing overtraining on a context-specific action, facilitating organization throughout training.

We will compare these agents of various levels of IBV to assess their impact on overall performance. In doing so, we plan to provide computational evidence that IBVs, such as SMAs, play a pivotal role in neuromotor adaptation.

## 2 SIMULATION

All simulations were run in Python, using PyBullet for the physics simulations and PyTorch for the neural network implementation (Greff et al., 2022; Paszke et al., 2017; Van Rossum & Drake, 2009).

### 2.1 ENVIRONMENT

The agent was placed in an empty environment with a floor. Depending on the task, one or more objects may appear at a specified distance from the agent and are regarded as targets that the agent must move toward. Gravity was set at $-9.8\,\mathrm{m/s^2}$ along the z-axis.

### 2.2 AGENT

Composed of four (4) joints, the agent resembles the human pointer finger fixed to the ground (see Figure 1). The joint closest to the ground is revolute, allowing for 360° rotation across the z-axis. The proceeding three (3) joints are hinges but constricted to rotate up to 90° across the y-axis. Each joint's angle of rotation, relative to the parent link, and current velocity serve as the inputs into the neural network architecture, which will, in turn, direct subsequent movements as its outputs.

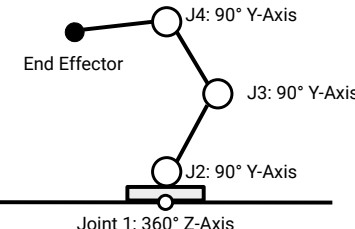

Figure 1: Agent schematic

### 2.3 NEURAL ARCHITECTURE

Using PyTorch, we initialize a fully interconnected feed-forward neural network composed of eight (8) inputs and eight (8) outputs corresponding to the four (4) joints and their respective velocities. The network included a single (1) hidden layer composed of rectified linear unit action function nodes. The number of nodes in the hidden layer were manually changed depending on the complexity of the experiment (see below). We trained the network on various tasks using unsupervised and supervised learning (see Algorithm 1). Backpropagation was optimized via Adaptive Moment Estimation (Kingma & Ba, 2017).

### 2.4 INTRINSIC BEHAVIORAL VARIABILITY MODEL

The goal of the Intrinsic Behavioral Variability (IBV) model was to inject variable behavior into the agent's representation, mirroring prenatal SMAs. Similar to evidence that prenatal SMAs initiate the human somatotopic representation within humans (Thomason et al., 2018), we designed the model for self-identification of the agent. We argue that this model, in building a representation of the self rather than of goal-directed motor actions, can provide variable behavior to the neural network while also mirroring the theorized neurological function of building somatotopic representations. Using unsupervised learning, backpropagating a loss value between the joint angle and velocity input and the joint angle and velocity output via a mean squared error, this model works to build representations of its motor system, correlating its desired outputs with its inputs. This framework is based on work also using unsupervised learning mechanisms to examine self-organization (Petersson et al., 2003).

### 2.5 REACHING MODEL

The goal of the Reaching model was to train the agent to reach a given target, mirroring the formation of ethological action representations through exploration of the environment. To do so, this model

trains the neural network using supervised learning, backpropagating a loss value between the motor output and inverse kinematics as a ground truth via a mean squared error. Supervised learning has been widely theorized to underlying motor learning (Kawato, 1990; Raymond & Medina, 2018).

---

**Algorithm 1** Agent Training

---

1: **function** TRAIN(params)
2:     env, robot ← InitializeSimulation(params.agent_type, params.box_type)
3:     brain ← CreateNeuralNetwork(input_size, params.hidden_size, output_size)
4:     **if** params.load_file exists **then**
5:         LoadWeights(brain, params.load_file)
6:     **end if**
7:     **for** time_step ← 1 **to** params.time_limit **do**
8:         current_state ← GetRobotState(robot)
9:         **if** params.train_type = 'IBV_Model' **then**
10:             output ← brain.ForwardPass(current_state)
11:             ApplyActions(robot, output)
12:             loss ← CalculateLoss(current_state, output)
13:         **else if** params.train_type = 'Reach_Model' **then**
14:             target ← CalculateInverseKinematics(robot, GetBoxPosition())
15:             loss ← TrainOnBatch(brain, current_state, target)
16:             output ← brain.ForwardPass(current_state)
17:             ApplyActions(robot, output)
18:             **if** CheckCollision(robot, box) **then**
19:                 **break**
20:             **end if**
21:         **end if**
22:         UpdateNetwork(brain, loss)
23:         SaveData(robot, loss, time_step)
24:         StepSimulation(env)
25:     **end for**
26:     SaveResults(brain, params.save_file)
27:     **return** time_step
28: **end function**

**Input:** agent_type: "robot_arm", box_type: "++", load_file: "previous_weights.pth",
        time_limit: 1000, train_type: "Reach_Model", hidden_size: 8,
        save_file: "simulation_results"
**Output:** final_step
29: params ← {agent_type, box_type, load_file, time_limit, train_type, hidden_size, save_file}
30: final_step ← Train(params)
31: **print** "Simulation completed at step:", final_step

---

Using the IBV and Reaching models interchangeably, we can mirror neuromotor development, training agents on a reaching task and intermittently training the agent on intrinsically variable behaviors, testing their performance across various novel internal and external stimuli.

## 3   EXPERIMENT 1: NOVEL SKILL LEARNING

From birth, humans are tasked with learning a plethora of new skills; we question how intrinsically variable behaviors may benefit motor skill learning. In this task, we looked at how well each agent performed when learning a novel skill.

### 3.1   TASK

For this task (see Figure 2), we used the Reaching model to train agents to reach for targets within the agent's vicinity. Over the first six hundred (600) epochs, agents reached for three (3) targets, each randomly selected to appear at a specific distance equidistant from the agent. Then, for two

hundred (200) epochs the agents continuously trained to reach for a novel target in a novel location also equidistant from the agent that it had never trained on prior. Finally, for two hundred (200) epochs, the agents returned to training on the original three (3) targets appearing randomly. We wanted to examine the agent's ability to learn a diverse array of skills, learn a novel one, and then relearn the prior skill. Each epoch allowed for a maximum of one thousand (1000) timesteps, ending earlier if the agent reached the target within the timestep window. For each epoch, at each timestep, the agents' end effector position, the loss error, and the neural network weight matrix were recorded.

For agents simulating IBV (i.e., H1 and H2), the agents would train on the IBV model prior to the start of training for one (1) extended epoch of 10,000 timesteps, to mimic the prenatal period. For H2, as this hypothesizes that IBVs encourage flexibility when repeatedly trained, every hundred (100) epochs of target-reaching included one (1) epoch of training on the IBV model, mimicking the postnatal period.

To assess possible performance differences between each agent, each agent was run twenty-five (25) times, using a different random seed.

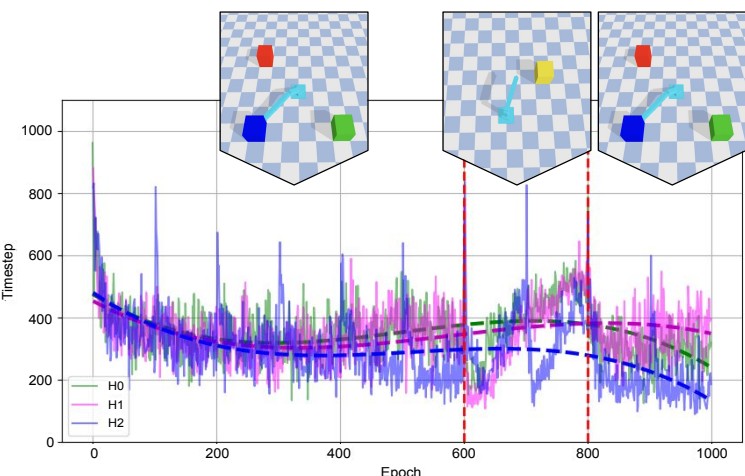

Figure 2: Experiment 1 behavioral performance.

## 3.2 ANALYSIS

To analyze possible differences between the performance of the agents, we looked at their average behavioral performance on the reaching task and their neural weight changes over twenty-five (25) runs for robustness.

To assess the agents' behavioral performance, we averaged each agent's performance (i.e., the number of timesteps it took for agents to reach the target) over the twenty-five (25) runs for each reaching epoch over time. We then compared the overall performance of the three (3) agents using ANOVA and a post-hoc Tukey's HSD test, if necessary.

To assess agents' neural network differences, we averaged the agents' weight matrices at each epoch of training. We then performed principal component analysis (PCA) on the matrices to reduce the dimensionality of the data for insight into neural weight changes in variability at various points in training. We compared agents' variability using ANOVA and the Mann-Whitney U Test.

## 3.3 RESULTS

For Experiment 1, we found a significant overall difference in behavioral performance between agents ($F(2, 2997) = 555.86, p = 4.74 \times 10^{-206}$). A post-hoc Tukey's HSD test showed significant differences between all agents when compared to one another.

We also found a significant difference in neural net weight variability between agents ($F(2, 72) = 16.56, p = 1.21 \times 10^{-6}$) (see Figure 3). Mann-Whitney U test showed that the Intermittent IBV

Hypothesis [H2] had a greater neural net weight variability when compared to the Training Hypothesis [H0] and Pre-Training IBV Hypothesis [H1] after initially training on the first three (3) targets ($U = 100, p = 3.90 \times 10^{-5}$ and $U = 83, p = 8.86 \times 10^{-6}$), after learning to reach the novel target ($U = 46, p = 2.45 \times 10^{-7}$ and $U = 43.00, p = 1.80 \times 10^{-7}$), and after returning to the three (3) targets ($U = 28, p = 3.58 \times 10^{-8}$ and $U = 45.00, p = 2.21 \times 10^{-7}$), respectively. There was an insignificant difference between H0 and H1.

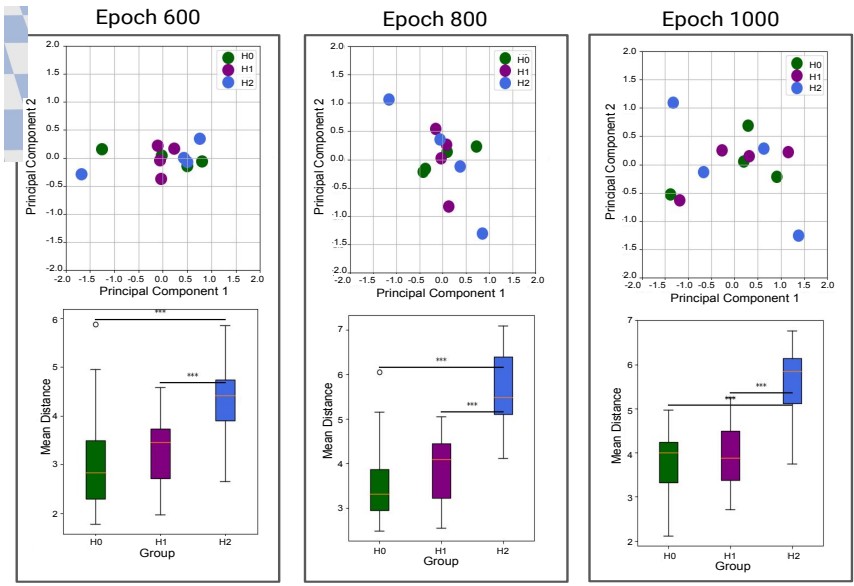

Figure 3: Experiment 1 neural performance.

## 3.4 SUMMARY

In this task, we looked to see how well each agent performed when learning a novel skill. We found that H2 outperformed H1 and H0 overall, learning the initial three targets, the novel target, and returning to the initial three targets faster than the others, evidence that interspersed behavioral variability improved performance and flexibility. In addition, we saw that H2 showed a higher rate of learned representational variability throughout training, evidence that intermittent IBV may increase exploration of the environment and flexibility of the neural network for novel learning and retention.

## 4 EXPERIMENT 2: AMPUTATION

Throughout development, the human motor systems change in numerous ways; in this experiment, we question how IBV may benefit adaptation to internal changes. In this task, we looked at how well each agent performed when adapting to amputation.

### 4.1 TASK

For this task (see Figure 4), we again used the Reaching model to train agents to reach for targets. After the initial six-hundred (600) epochs of reaching for three (3) targets randomly, however, we removed the third link and joint of the agents, increasing its overall size to compensate for its loss link. The agents were then tasked with reaching for the same three (3) targets over another six hundred (600) epochs.

Each agent was run twenty-five (25) times.

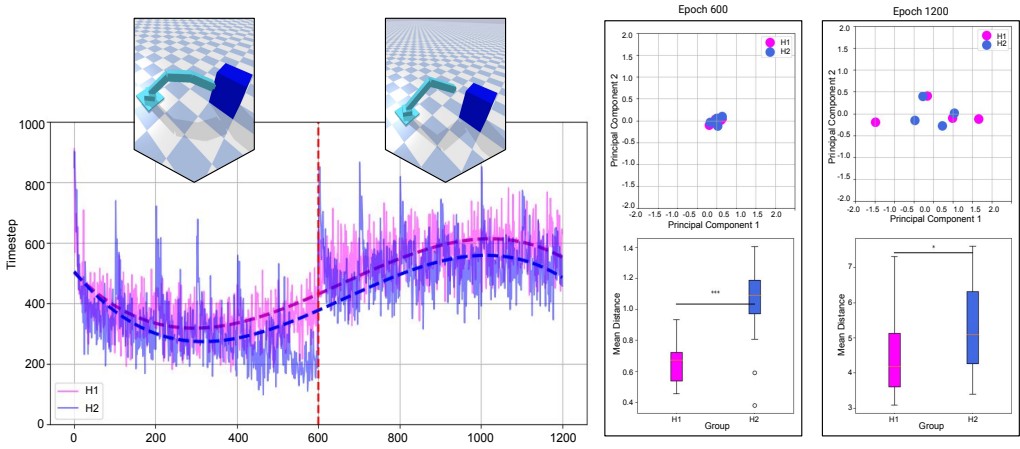

Figure 4: Experiment 2 behavioral and neural performance.

## 4.2 RESULTS

For this study, we chose to look at the Pre-Training IBV Hypothesis [H1] and the Intermittent IBV Hypothesis [H2] exclusively. We chose this approach to emphasize the neuromotor developmental evidence that IBVs are essential to representation initialization within humans (Blumberg et al., 2013b). In addition, Experiment 1 gave strong indication that the Pre-Training IBV Hypothesis [H1] would mirror H0's results. We found a significant overall difference in performance between H2 and H1 $(F(1, 2400) = 116.76, p = 1.31 \times 10^{-26})$. We also found that H2 had a higher rate of representational variability pre- $(F(1, 48) = 62.52, p = 3.04 \times 10^{-10})$ and post- $(F(1, 48) = 4.57, p = 0.0376)$ the amputation than H1.

## 4.3 SUMMARY

In this task, we looked to see how well each agent performed when adapting to an amputation. We found that H2 outperformed H1 overall, adapting faster to the physiological changes, evidence that interspersed behavioral variability improves performance and flexibility to morphological changes. In addition, we saw that H2 showed a higher rate of representational variability pre- and post-amputation, evidence that intermittent IBV may increase the flexibility of the agent's representation for adaptation to internal changes.

## 5 EXPERIMENT 3: NEURAL STROKE

Whether through neurogenesis or stroke, our brains can change quite dramatically over time. In this experiment, we examined how IBV may improve an agent's adaptation to such representational dramatic changes. In this task, we looked at how well each hypothesis performed when adapting to neural network knockouts.

### 5.1 TASK

For this task (see Figure 5), rather than augmenting the physiological state of the agents, we choose to restrict their neurological capacity. After the initial six hundred (600) epochs of reaching for three (3) targets, we silenced a single hidden neural node from the trained eight (8) node neural network, effectively simulating a neural stroke. The agents were then tasked with reaching for the same three (3) targets over another three-thousand (3000) epochs, due to the severity of the augmentation.

Each agent was run twenty-five (25) times.

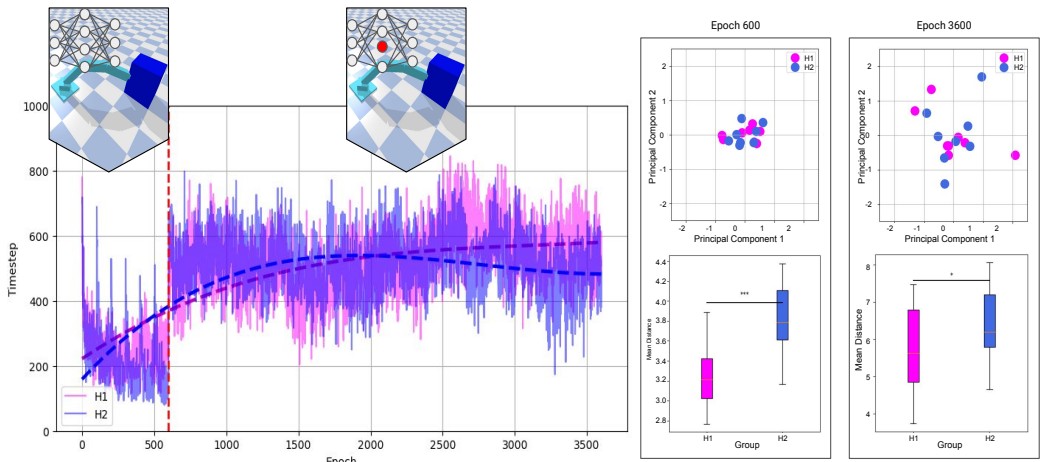

Figure 5: Experiment 3 behavioral and neural performance.

## 5.2 RESULTS

We found a significant overall difference in performance between the Intermittent IBV Hypothesis [H2] and the Pre-Training IBV Hypothesis [H1] ($F(1, 7198) = 56.97, p = 4.98 \times 10^{-14}$). We also found that H2 had a higher rate of representational variability pre- ($F(1, 48) = 38.40, p = 1.25 \times 10^{-7}$) and post- ($F(1, 48) = 5.18, p = 0.0274$) the neural stroke than H1.

## 5.3 SUMMARY

In this task, we looked to see how well each agent performed when adapting to a neural network knockout. We found that H2 outperformed H1 overall, adapting faster to the representational changes, evidence that interspersed behavioral variability improves performance and flexibility to representational changes. In addition, we saw that H2 showed a higher rate of representational variability pre- and post-stroke, evidence that intermittent IBV may increase the flexibility of the neural network for adaptation to severe internal changes.

## 6 DISCUSSION

In this study, we examined the impact intrinsic behavioral variability plays in facilitating a flexible and dynamic representation. Across three (3) experiments we tested agents of varying behavioral variability, examining the influence IBV may have on overall behavioral performance while simultaneously observing the underlying neural representational changes involved. In Experiment 1, we tested the influence of IBV in adapting to external changes: in learning a novel motor skill. We found that agents with intermittent IBV outperformed those with less persistent IBV, evidence that persistent IBV may facilitate a representation more flexible to learning novel motor tasks. This is especially true when observing each agent's performance when learning to reach the novel target and then returning to the original three (3) targets. In the Intermittent IBV Hypothesis [H2]'s behavioral performance (see Figure 2), there were distinct spikes in performance, most likely due to the intermittent IBV training. When compared to its significantly higher representational variability, it can be argued that the IBV, though in the short term reduced performance, encouraged broader exploration of the environment, increasing overall performance.

In Experiments 2 and 3, we tested the influences of IBV in adapting to amputation and neural deficit. We again found H2 outperforming its counterparts. Network analyses showed higher rates of representational variability in H2 pre- and post-deficit, evidence that H1 may have overfitted to a suboptimal solution while H2's continued IBV encouraged the discovery of more optimal (i.e., faster) solutions. We note, however, that though H2 performance outpaced H1, the neural weight variability by the end of training between both agents were more closely similar. It can be argued

that H2's persistent IBV training gave it a behavioral advantage throughout but, with time, enough training on a task could elicit similar neural weight variability and solution discovery. Overall, we showed strong computational evidence that IBV plays a fundamental role in facilitating a motor representation by encouraging adaptation to behavioral, physiological, and neurological changes, supporting neuromotor developmental theory.

When taken in the context of human embodied development, this paper's results align strongly with our evolving understanding of human neuromotor development. Numerous papers have begun to show evidence that the motor cortex's representation of the body is not only more dynamic but also more complex than previously imagined (Berlot et al., 2020; Ejaz et al., 2015; Gordon et al., 2023; Graziano, 2016; Makin & Krakauer, 2023). The organization, and developmental trajectory, however, have yet to be fully described. Our work seeks to formalize this representational trajectory. We argue that, prenatally, IBVs such as SMAs serve to initiate a representation of the body, as argued by Blumberg and colleagues (Thomason et al., 2018). This representation postnatally, however, has been shown to change dramatically (Dooley & Blumberg, 2018). As actions are learned and incorporated within the motor cortex (Graziano, 2016), these same SMAs, now present during postnatal rest (Sokoloff et al., 2020), serve as a means of "resetting" our representation towards its initial state.

The interweaving of action representations and self-organizing representations prevent any singular actions from being prioritized within the representation, encouraging a more flexible and dynamic mapping. This would support evidence that behavioral entrenchment in an action does not necessarily alter neural representations of the individual motor units (Beukema et al., 2019; Beukema & Verstynen, 2018). Heavily trained actions become localized rather than globalized. In Experiment 1, simulations with intermittent IBV not only learned novel tasks faster, but also recovered previously learned skills faster, evidence in favor of SMAs playing a role in maintaining prior skill knowledge, while being able to efficiently learn new ones. Experiments 2 and 3 further support this claim, showing better adaptation to ingrained representations of the agent's internal configurations when trained with IBV. Together, this work provides computational evidence that SMAs play a fundamental role in facilitating a motor representation of both the body's form and function, encouraging adaptation throughout development.

This work also aligns strongly with the ongoing developmental robotics literature on action generation. Another form of SMA, motor babbling, the seemingly incoherent attempt at vocalization by infants is a commonly studied phenomenon in robotics (Aoki et al., 2016; Bullock et al., 1993; Caligiore et al., 2008; Mahoor et al., 2017). Researchers have used the concept, of eliciting random movements within robots, to build an inverse kinematics model of its environment. These same models have been used for self-modeling; and when iteratively used, have been shown to allow adaptation to physiological augmentation (Bongard et al., 2006).

One compelling aspect of Experiment 1's task design was the potential for agents to recall previously learned skills after learning a novel one. The Intermittent IBV Hypothesis [H2] appeared to overcome this issue of catastrophic interference, relearning and improving upon the initial task faster than its counterparts. This result speaks to evidence within both neuroscience and connectionist theory that the brain may form distributed representations for context-specific situations (Ellefsen et al., 2015; McCloskey & Cohen, 1989; Plas et al., 2024), enabling flexible learning, retention, and maintenance. Future questions to be explored include the impact of training schedules on the representational efficiency of low-resourced agents.

A question that arises from this work, however, is whether we can consider IBV as simply noise in the system. Research has shown that noise can sometimes elicit better long-term performance by preventing overfitting (Gupta & Gupta, 2019). Though we include a supplemental experiment (see Appendix: Figure 6) showing the Intermittent IBV Hypothesis [H2] outperforming the Training Hypothesis [H0] with noise injected into the network ($p < 0.05$), we do not neglect the notion that noise may play a role in human motor learning and adaptation. In hindsight, we consider IBV such as SMAs as a form of noise within the motor cortex, degrading representations as to prevent overtraining. In future work, we plan to scale these simulations, exploring various augmentations such as ecologically-relevant gradual growth trajectories, and mapping behaviors to specific neural weight representations.

## 7 CONCLUSION

How we represent our dynamic bodies defines how we interact with the dynamic world around us. In this paper, we argue that our motor representations are just as dynamic, exploring the influence intrinsic behavioral variability such as spontaneous muscle activations have in motor representations. Across three experiments, we show evidence that intrinsic behavioral variability facilitates the initialization, maintenance, and updating of agents' representation by encouraging adaptation to behavioral, physiological, and neurological changes. Taken together, this work provides a biologically plausible computational framework for understanding the neurological and behavioral effects intrinsic behavioral variability may have on the manifestation of human neuromotor adaptation.

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

## A  APPENDIX

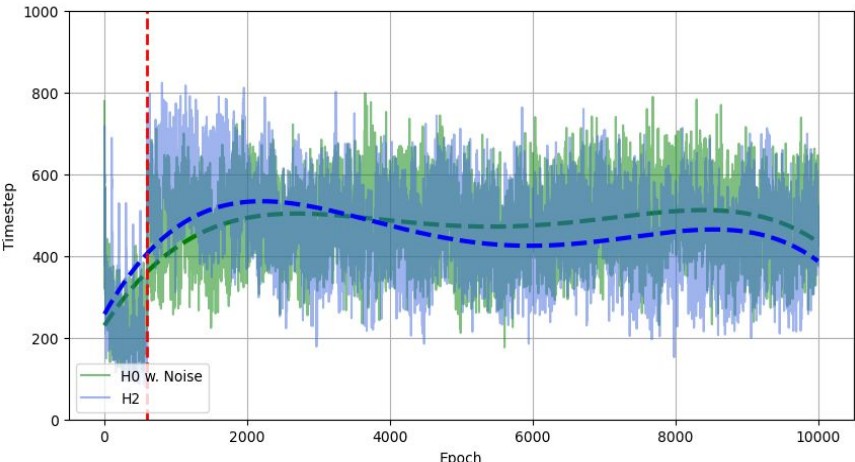

Figure 6: Supplemental experiment.

