# OpenReview forum: "Intrinsic Behavioral Variability Facilitates Flexible Representations: A Neuromotor Developmental Perspective"
_ICLR.cc/2025/Conference — ICLR 2025 Conference Withdrawn Submission_

### Official Review · Reviewer_izSg · 2024-10-31

**Soundness:** 1
**Presentation:** 1
**Contribution:** 1
**Rating:** 3
**Confidence:** 5

**Summary:**

This paper proposes that intrinsic behavioral variability (IBV), inspired by spontaneous muscle activations, facilitates the development of flexible representations for neuromotor control. Through three experiments—learning novel skills, adapting to amputation, and compensating for neural deficits—the authors examine whether IBV improves network adaptability.

**Strengths:**

- The idea of using variability inspired by SMAs has merit and aligns with biological findings on the developmental role of variability in motor learning.

**Weaknesses:**

- The training setup is vague. It’s unclear whether the network is trained with supervised or unsupervised learning, whether it learns a full action sequence or only a final position, or what constitutes the "10'000-time-step pre-training." The episodic framework lacks clarity on fundamental aspects, including definitions of "epochs" and "episodes", as well as termination criteria.
- Key aspects of the experiments are poorly explained. The "mean distance" metric is ambiguous, using cosine similarity on the neural weights could offer more direct insight.
- No comparison with other exploration or variability mechanisms (e.g., random noise or dropout) is provided. Without baselines, it’s impossible to determine whether IBV confers any meaningful advantage.
- The principal component trajectories of H1 and H2 are inconsistent across experiments. In Experiment 1, H1 and H2 differ considerably, but they appear similar in later analyses. This discrepancy, combined with the lack of explanation, suggests potential issues with experimental consistency. Further, H1’s lack of initial performance advantage, despite pre-training, raises doubts about the claimed benefits of IBV. There is also a very small performance difference between the hypotheses.
- The paper implies a continual learning setting but does not demonstrate any such progression. Ideally, the model’s performance would initially decrease with task changes and improve afterwards. However, in Figure 2, performance appears stable, even decreasing over time.
- In the neural stroke experiment, silencing a single node leads to significant degradation in performance. This raises questions about model resilience and whether a control network with 7 nodes (trained from scratch) would achieve similar results, clarifying whether other factors are at play.
- Overall, the framework and experiments are not sufficient to support the paper’s conclusions.

**Questions:**

- What exactly constitutes “intrinsic variability"? This is a central concept but it is never properly defined, leaving ambiguity about what variability means within the training context. Specifically, how does the "pre-training" process incorporate variability? What does “10'000 time-step pre-training” involve?
- If training is supervised, what is the target and how is the loss computed?
- Why doesn’t H1 demonstrate performance advantages from pre-training?

---

### Official Review · Reviewer_zeDu · 2024-11-03

**Soundness:** 2
**Presentation:** 3
**Contribution:** 1
**Rating:** 3
**Confidence:** 4

**Summary:**

The papers discusses development and robustness of motor representations from a biological perspective. Inspired by prenatal twitches in human development, the authors design a learning algorithms and test it in a robot arm. The authors “question” whether a similar learning scheme could be present in life-long motor adaptation. Three experimental situations are featuring simulated reaching tasks under various changes, showing that behavioral variability can be  effective in motor learning.

**Strengths:**

The different hypotheses provide a useful approach (although comparability needs to be assessed more strictly) to the problem, which in robot learning discussions could be even more generally formulated.

Taking inspiration from biology can be a useful approach to complex problems in robot learning.

The paper is clear and well written.

**Weaknesses:**

The statement in the abstract: “Dynamic human movement necessitates a dynamic representation of the body.” should be reformulated and evidenced in some way. For example, a frictionless double pendulum (as an oversimplified model of human arms or legs) produces all kinds of dynamic movement without any representation. Likewise the discussions of representations in psychology may be seen as still having to reach a satisfying conclusion, but clearly show that the subject is more complicated. (Compare also the less decisive statement in the conclusion: “we argue that our motor representations are just as dynamic …”).

Consider using the word “index finger” instead of “pointer finger”. Also “ethological action” is not a common expression in the present context (i.e. avoid or explain). Wordings like these are sometimes considered as evidence that LLM have been used in the generation of the text.

The model is not realistic. Fingers do not have a 360 degrees rotation at the base. Wrist rotation is usually up to 180 degrees, but wrist movement it not even mentioned (and would require another hinge-type joint at the base). Also, the arm moments will be more important in the early formation of motor representations than those of a single finger.

SMA is known as “supplementary motor area”, and using this abbreviation differently can lead to confusion.

**Questions:**

What does the current work add conceptually to existing work on motor babbling? How does it improve the biological realism of such claims? What does it add in terms of benefits for robot learning were various exploration schemes were adopted from machine-learning?

I'm not sure whether I understood experiment I correctly. It is assumed the block grasping task is representative for prenatal motor learning?

Significance levels are useful under “natural noise” conditions. If the noise is computer-generated, it is necessary to add some procedure to gauge the noise, or to show how the significance scales with number of experiments, or similar. It should at least be noted that obtained significance level do not translate into biological nor robotic significance. The question is: Can you give an interpretation of the significance level within the present context and within the interdisciplinary context?

**Details Of Ethics Concerns:**

Experimental amputation and induced strokes are used in simulations with an explicit biological interpretation. As this is not further discussed, it could seem as if such procedures were generally recommended for biological experiments, although there they can be used, if at all, only in exceptional cases.

I don't think "dual submission" belongs here, because, if I were checking this, I would most likely destroy the anonymity of the authors which is prohibited by the code of conduct. However, accepting this option as applicable, I would need to flag it, because it is *possible* that the paper was submitted also elsewhere, just like I would flag a paper where it is *possible* that any participants were harmed. In other words, this should be checked at a later stage of the review process, if at all.

---

### Official Review · Reviewer_mYMG · 2024-11-03

**Soundness:** 2
**Presentation:** 1
**Contribution:** 1
**Rating:** 3
**Confidence:** 4

**Summary:**

This work investigates the role of "intrinsic behavioral variability" (IBV) for motor learning, which seems to be a form of multi-task learning in which a primary supervised objective (for robot arm reaching) is interleaved with a secondary unsupervised objective (for next-state prediction). The agent is a PyBullet physics simulation of a 4 joint robot arm, which aims to perform a reach-to-target task under 3 experimental setups: (1) "novel skill learning", i.e. in the presence of a new target location during training, (2) "amputation", i.e. in the presence of a morphological perturbation during training, (3) "neural stroke", i.e. in the presence of a neural activity perturbation during training. Comparing results for 3 hypothesized learning algorithms (supervised only, unsupervised pretraining, supervised/unsupervised interleaved), the work makes the case for the interleaved algorithm.

**Strengths:**

**Significance/Motivation**: The work adopts an interesting approach by using computational modeling on an embodied task to provide insights on brain theories. The motivation is reasonably sound, since it's generally accepted that the brain uses a variety of learning algorithms (combining unsupervised and supervised rules) to learn a body model and use that model to execute specific tasks.

**Weaknesses:**

Unfortunately, I think the weaknesses outweigh the strengths in this manuscript as it currently stands. I hope the following feedback will be useful for improvements.

---

**Clarity**: The manuscript is quite challenging to read, due to vague language, missing technical details, and stylistic quirks. In particular:

**W1: Vague/fuzzy language.** The core concept "intrinsic behavioral variability" is not clearly defined. Is it a model, an algorithm, a task curriculum? How exactly is it related to twitches, as there are no twitches in the model? Other phrases like "representational maintenance and flexibility" (071), "we hypothesize that IBV facilitates a flexible representation of an agent's behavioral repertoire and self" (076) are also hard to interpret concretely and seem rather light on content.

**W2: Missing equations.** The key losses are defined in words only:
"backpropagating a loss value between the joint angle and velocity input and the joint angle and velocity output via mean squared error" (IBV unsupervised loss),
"backpropagating a loss value between the motor output and inverse kinematics as a ground truth via a mean squared error" (reaching supervised loss).
I couldn't understand what this means. In particular, is the IBV loss using the current state $o_t$ to predict the next state $o_{t+1}$ or the agent actions $a_t$? It seems like the latter. Also, is `current_state` is changing between algorithm lines 10 and 12 (after applying the action on line 11)?

**W3: Observation/action space.** What exactly are they? The manuscript only says:
"Each joint's angle of rotation, relative to the parent link, and current velocity serve as the inputs into the neural network architecture, which will, in turn, direct subsequent movements as its outputs." What specifically (in math) are the actions? Joint torques, velocity, positions etc.? If the latter, do you use a PD controller to convert into torques for PyBullet?

**W4: Figures.** Every figure is missing substantive captions. They use small fonts. There's no context in Appendix, just a figure with the caption "Supplemental experiment."

**W5: Numbers.** There's a weird stylistic quirk of writing out every number in words then adding a parenthesized Arabic numeral (e.g. "one thousand (1000)", "one (1)", "two (2)", "three (3)", …). It is superfluous and rather annoying/distracting to read… please remove and keep only the Arabic numeral!

**W6: Algorithm.** The algorithm is provided in too low-level detail, like a less readable version of code. Please keep algorithm to high level.

**W7: Hypotheses.** It would be clearer to use informative names like "Supervised Only", "Pretrained" rather than "H0", "H1", etc.

---

**Quality**: Given the lack of manuscript clarity, it's hard to assess the technical quality. But some main questions:

**W8: Spiky graphs.** Why are the graphs so spiky? Shouldn't they be an average of multiple random seeds? How many random seeds are used? What are the dashed lines (averages over seed, or a smoothed version of the line)?

**W9: Small effect sizes.** The training curves seem basically overlapping. There are p-values that have been calculated to suggest statistically significant differences, but the effect sizes seem small at best. Especially if the curves are a single seed, it doesn't seem like a compelling case for H2, given variance in task difficulty between episodes.

**Questions:**

See weaknesses.

---

### Official Review · Reviewer_VLMG · 2024-11-06

**Soundness:** 1
**Presentation:** 2
**Contribution:** 1
**Rating:** 1
**Confidence:** 4

**Summary:**

The paper examines the role of spontaneous muscle twitches in shaping neural representations. This phenomenon is explored through a simple model in which a 4 DoF arm, controlled by a feedforward neural network, performs a reaching task. Muscle twitches are hypothesized to foster self-identification in the agent, modeled by training the network to match its outputs to its inputs. The primary task involves supervised learning of an action sequence derived from inverse kinematics, with the aim of assessing how self-identification impacts performance and robustness in reaching.

**Strengths:**

The paper presents interesting biological phenomena in an accessible way.

**Weaknesses:**

* Oversimplified modeling:
While toy models can be useful for generating clear, tractable insights, this one falls short of effectively capturing the biological processes in question.
The muscle twitch model is limited to learning an identity function, training outputs to match inputs without dimensionality reduction (line 195). Although this procedure may act as a form of regularization, it doesn’t align with the biological phenomenon of self-identification because learning the identity function here requires access to the "ground truth" arm configuration. In a biologically realistic system, self-identification would rely on internal state estimation, as the brain lacks direct access to effector positions.

* Unconvincing performance and statistical interpretation:
The conclusion are based on statistical significance between the agents subject to different training, with two main limitations.
First, treating each epoch as an independent sample inflates the sample size, resulting in extremely small p-values (e.g., p = 1e-206), which overstates the statistical significance.
Second, the emphasis should be on the performance improvement. Here, the effect size appears modest, and robustness claims are weak—for example, performance does not recover from the ablation of a single neuron (Fig. 5).

**Questions:**

* Explain how the loss of the IBV model would be computed by the brain.

* Clarify contribution to the field of motor babbling.

---

### Note · Authors · 2024-11-26

I have read and agree with the venue's withdrawal policy on behalf of myself and my co-authors.